# The development and validation of an Inhomogeneous Wind Scheme for Urban Street (IWSUS-v1)

Zhenxin Liu[*1],Yuanhao Chen[1],Yuhang Wang[2],Cheng Liu[3],Shuhua Liu[4],Hong Liao[1]

1 Jiangsu Key Laboratory of Atmospheric Environment Monitoring and Pollution Control/Jiangsu Collaborative Innovation Center of Atmospheric Environment and Equipment Technology, School of Environmental Science and Engineering, Nanjing University of Information Science and Technology (NUIST), Nanjing 210044, China
2 School of Earth and Atmospheric Sciences, Georgia Institute of Technology, Atlanta, GA, USA
3 Jiangxi Province Key Laboratory of the Causes and Control of Atmospheric Pollution/School of Water Resources and Environmental Engineering, East China University of Technology, Nanchang 330013, China
4 Department of Atmospheric and Oceanic Sciences, School of Physics, Peking University, Beijing, China

*Correspondence* to: Zhenxin Liu ( liuzhenxin@nuist.edu.cn )

**Abstract.** The layout of urban buildings shows significant heterogeneity, which leads to the significant spatial inhomogeneity of the wind field in and over the canopy of urban street canyons. However, most of the current urban canopy models do not fully consider the heterogeneity of the urban canopy. Large discrepancies thus exist between the wind speeds simulated by the current urban canopy models and those observed in the street canyon. In this study, a parameterization scheme for wind field, Inhomogeneous Wind Scheme for Urban Street (IWSUS), is developed to better characterize the heterogeneity of the urban canopy. We use a Computational Fluid Dynamics method to generate the IWSUS scheme and compare with observations of wind profile and turbulent flux in and over the street canyon for validation. In IWSUS, the wind speed vertical profiles at six representative positions located in a typical street canyon (i.e., the windward or leeward side of a long straight street, or the inflow or outflow end) are parameterized separately. The wind profile by IWSUS thus can better describe the horizontal heterogeneity of the urban near-surface wind field, e.g., the dynamic drag effect of building in the lower atmosphere layer over the urbanized land use. The validation based on observations shows that the performance of simulation results by IWSUS is better than that by logarithmic-exponential (exp-log) law widely used in the current urban schemes. We consider typical building arrangement and specific street orientations in IWSUS for wind field simulations, which can better match the distribution characteristics of street canyons around the observation point in the street canyon. The averaged wind profiles and turbulence energy fluxes in the model grids of urban area by IWSUS are also nearer to the observation than those by exp-log law. The normalized mean errors (NME) between the simulated and the observed vertical average wind speed are 49.0% for IWSUS and 56.1% for exp-log law in the range from the ground to four times the average height of the building, and 70% for IWSUS and 285.8% for exp-log law in the street canyon (range from the ground to building top). This study proves that the accuracy of land surface process and near-grounded meteorological process simulations over urban canopy can be improved by fully considering the heterogeneity of the urban canopy layout structures and the inhomogeneous of wind field distributions in and over the street canyon. The IWSUS is expected to be coupled with mesoscale atmospheric models to improve the accuracy of the wind field, land surface energy budget, meteorological, and atmospheric chemistry simulations.

# 1 Introduction

The wind speed in planetary boundary layer (PBL) is one of the key meteorological factors affecting the land surface energy balance, regional weather and climate, and also the distribution of near surface atmospheric pollutants (Xu et al., 2015; Deng et al., 2018; Peng et al., 2021). Urbanization causes complex canopy structures based on the building-street canyon unit, which are quite different from the natural vegetation-soil type canopy (Ren et al., 2015; Zhu et al., 2017; Paul et al., 2018; Colter et al., 2019). The complex shapes and arrangement of urban buildings layout lead to the great complexity on the

distributions of wind and turbulence field and show obvious spatial inhomogeneity of the three-dimensional wind field in and over the street canyons (Li et al., 2018; Li et al., 2020). The inhomogeneous wind field in and over the urban street canyon not only leads to quite complexed characteristics of momentum and mass fluxes over the urbanized urban land cover (Ryu and Baik, 2013; Ganbat et al., 2014a; Ganbat et al., 2014b), but also causes the significant spatial variation of air pollutant concentration in urban street canyons (Haman et al., 2012; Banks et al., 2015; Miao et al., 2015; Huang et al., 2017; Jaenicke

et al., 2017; Miao and Liu, 2019; De Arruda Moreira et al., 2020; Yang et al., 2021).

  The wind parameterization schemes for urban area applied in most of the current air quality models have not fully considered the heterogeneity of the urban canopy layout structures, thus the inhomogeneity of the wind field in and over the urban canopy have not been fully considered, either (Masson, 2000; Kusaka et al., 2001; Kadaverugu et al., 2019; Masson et al., 2020). The classical logarithmic-exponential (log-exp) profile schemes widely applied in those simulations is based on the

hypothesis of homogeneity in horizontal (Masson, 2000; Martilli, 2002), which produces the averaged wind field in the model grid points of urban land surface. But it is unreasonable to directly compare the wind field simulated by exp-log law with the observations, since the observation spots located in the urban street canyon are affected by the inhomogeneous air flow field, caused by the layout distribution of surrounding buildings. The measurement experiments set up (Sha et al., 2021) in Nanjing showed the variations of observed surface energy flux in different locations over the urban canopy, which proves the

heterogeneity of energy flux over the different representative positions of urban canopy, e.g., on the tops of building roof or street canyon. The inconsistency of spatial representativeness between the point observation and the model grid simulation results leads to significant differences in their numerical values. This deviations between the simulations and observations in and over the urban canopy can be found in both the meteorological and chemical processes modeling studies, e.g., intensity of urban heat island (UHI) and nocturnal stability properties of PBL(Nair, 2011; Kusaka et al., 2012; Schubert, 2014; Miao et al.,

2015; Ren et al., 2019a; Ren et al., 2019b). Husain et al. showed that the Town Energy Budget (TEB) scheme produced a lower daytime maximum and a higher nighttime minimum for temperature simulation in urban region and overestimated UHI during the late evening hours (Husain et al., 2013). Thus the near land ground pollutant concentrations during the heavy air pollution events in simulations are overestimated in urban area (Fallmann et al., 2016; Santiago et al., 2020; Xue et al., 2021). Studies suggest that these simulation error of near-surface pollution concentration mainly comes from the inaccurate simulation

of near-surface atmospheric environmental capacity by the model, which is due to the fact that the model cannot well simulate

the energy distribution and dynamic characteristics of urban surface canopy.(Saiz-Lopez et al., 2007; Baklanov et al., 2011; Harrison, 2018; Khalil, 2018; Chen et al., 2020).

In summary, a scheme that fully considers the heterogeneity of wind field caused by the heterogeneous urban canopy structure is very important for further accurate simulation of meteorological processes (Sützl et al., 2020), the surface fluxes (Sha et al., 2021), the PBL atmospheric conditions (Miao et al., 2009; Thatcher and Hurley, 2011; Miao and Chen, 2014), and also the air pollution in and around the urban regions (Liu et al., 2020; Liu et al., 2021).

In the present study, we developed an Inhomogeneous Wind Scheme for Urban Street (IWSUS) to fully describe the heterogeneous characteristics of the wind field in and over the urban canopy. Firstly, a typical building-street layout model was constructed by using Computational Fluid Dynamics (CFD) method, and the three-dimensional wind fields inside and above the street canyon were solved under the initial and boundary conditions of different background wind directions and street canyon geometric parameters. Then the characteristic vertical wind profiles were parameterized over six representative positions in the typical street canyon and finally summarized to be the IWSUS scheme. Moreover, the wind profiles simulations by IWSUS were verified and corrected by using the wind profile observation data in a street canyon. The validations of IWSUS were also set up by comparing observed wind vertical profiles and surface fluxes with those calculated based on IWSUS.

The brief introductions of the observations, the CFD methods can be found in Section 2.1 and 2.2; the initialization settings of the scheme and the introduction of regression method are described in Sect.2.3. The parameterization results and discussions are in Sect. 3; the validation of IWSUS is in Sect. 4. Then the conclusions are given in Sect. 5.

## 2 Data, Methodology and Model

### 2.1 CFD Model description

The CFD method was applied in study to analyze how the geometric characteristics of the street canyon influence the wind field in and over the urban canopy. Series of numerical experiments were set up in the street canyon scenario by the CFD model, OpenFOAM-v8 (https://openfoam.org/ ), in which the average street width and building height were set as the control variables. The Naïve-Stokes equations with the standard $k - \varepsilon$ turbulence closure scheme was solved by the following equations, and the wind velocity and turbulence vertical profiles in the six special locations were obtained:

$$\frac{\partial u_i}{\partial x_i} = 0 \tag{1}$$

$$\frac{\partial u_i}{\partial t} + \frac{\partial}{\partial x_j}\left(u_j u_i\right) - \frac{\partial}{\partial x_j}\left[\nu_{\text{eff}}\left(\frac{\partial u_i}{\partial x_j} + \frac{\partial u_j}{\partial x_i}\right)\right] = -\frac{\partial p^*}{\partial x_i} \tag{2}$$

$$\frac{\partial k}{\partial t} + \frac{\partial(k u_i)}{\partial x_i} = \frac{1}{\rho}\frac{\partial}{\partial x_i}\left[\frac{\nu_t}{\sigma_k}\frac{\partial k}{\partial x_i}\right] + \frac{\nu_t}{\rho}\left(\frac{\partial u_i}{\partial x_j} + \frac{\partial u_j}{\partial x_i}\right)\frac{\partial u_j}{\partial x_i} - \varepsilon \tag{3}$$

$$\frac{\partial \varepsilon}{\partial t} + \frac{\partial(u_i \varepsilon)}{\partial x_i} = \frac{1}{\rho}\frac{\partial}{\partial x_i}\left[\frac{\nu_t}{\sigma_\varepsilon}\frac{\partial \varepsilon}{\partial x_i}\right] + \frac{c_1 \nu_t}{\rho}\frac{\varepsilon}{k}\left(\frac{\partial u_j}{\partial x_i} + \frac{\partial u_i}{\partial x_j}\right)\frac{\partial u_j}{\partial x_i} - C_2 \frac{\varepsilon^2}{k} \tag{4}$$

where $u_i$ is the $i$th component of wind velocity, $p^*$ is the modified mean kinematic pressure, $\nu_{\text{eff}}$ is the effective kinematic viscosity which is expressed as $\nu_{\text{eff}} = \nu + \nu_t$ with $\nu$ for molecular viscosity, $\nu_t$ is turbulence viscosity expressed as $\nu_t = \rho C_\mu k^2/\varepsilon$, $k$ represents turbulent kinematic energy, and $\varepsilon$ is turbulent dissipation. Meanwhile, $k - \varepsilon$ model contains five empirical constants in Eq.(3) and Eq.(4), and the values are listed as follows: $C_\mu = 0.09$; $C_1 = 1.44$; $C_2 = 1.92$; $\sigma_k = 1.0$; $\sigma_\varepsilon = 1.3$.

## 2.2 Simulation initial setting up

A typical geometric structure for a basic unit of urban street-building canyon was set up in OpenFOAM-v8 model. As shown in Figure 1a, the average height of building (defined as H for short), the width (W) and orientation of the street canyon are the key parameters to define the structure. We set length of buildings, width of buildings, and width of streets as constants in this research, which were 30 m, 5 m, and 20 m, respectively. The aspect ratio (AR) factor, defined as the ratio of building heights (H) to street width (W), was used to represent the degree of urban development. AR is a key factor effects the wind fields in and over the street canyon, as well as the background wind speed. We conducted numerical sensitivity simulations for wind speed and AR factor.

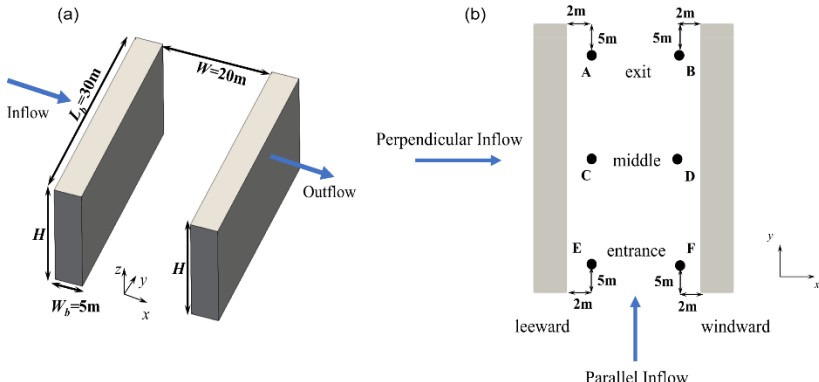

**Figure 1 The conceptual figure of a typical street canyon and its geometrical parameters initialization. Figure 1a: the blue arrows denote the inflow perpendicular to street canyon; Figure 1b: Schematic diagram of six representative positions in street canyon. A, C, E are in the leeward side and B, D, F in the windward. For oblique inflow, E, F are in the entrance side, A, B in the exit side, and C, D in the middle of the street canyon; the vertical wind profiles over these representative positions are considered to be different from each other.**

As the direction of the background wind also influences the wind within street canyon, we initialized the background wind speed in simulation as the inflow perpendicular ($u$ in $x$ direction) to the orientation of street canyon. The wind speed was initialized to be 0.1m/s, 1m/s, or 8m/s as the inlet boundary condition in OpenFOAM in different scenarios. The k and $\varepsilon$ in the inflow boundary are determined by $k = 1.5(I\overline{u_0})^2$ and $\varepsilon = C_\mu^{3/4}k^{3/2}/l$ respectively where $I$ is the turbulent strength and $l$ is the turbulent characteristic length scale. The no-Slip boundary is applied in the ground and building walls, and the other

boundaries are all set to be zero-gradient. The settings of boundary conditions in this study are widely used in urban street CFD simulations which is validated by wind tunnel experiments (Ai and Mak, 2017, 2018; Huang et al., 2019; Mirzaei, 2021)..All the initialization setting for every scenario set in CFD cases are summarized in Table 1. The domain size $L_x \times L_y \times L_z$ is 200m × 100m × 50m. The grid resolution in CFD modelling were set to be 1m in all scenarios. The direction of street orientated, lateral, and vertical are marked as $y$, $x$, and $z$, respectively. In meso-scale atmospheric models, the lowest layer of free atmosphere is often set as about 1.5-2 times the average height of urban canopy. Since IWSUS will be mainly applied in the meso-scale models, the vertical height of the domain is set as 50 m in our CFD experiments to match with the requirements in meso-scale model.

**Table 1 initial parameters set for a typical street canyon modeled in OpenFOAM.**

| Scenario group 1: inflow perpendicular to street | | Scenario group 2: inflow parallel to street | |
|---|---|---|---|
| Inflow (m/s) | AR | Inflow (m/s) | AR |
| | 0.25 | | 0.25 |
| | 0.5 | | 0.5 |
| $u=0.1$ | 1 | $v=0.1$ | 1 |
| | 1.5 | | 1.5 |
| | 0.25 | | 0.25 |
| | 0.5 | | 0.5 |
| $u=1$ | 1 | $v=1$ | 1 |
| | 1.5 | | 1.5 |
| | 0.25 | | 0.25 |
| | 0.5 | | 0.5 |
| $u=8$ | 1 | $v=8$ | 1 |
| | 1.5 | | 1.5 |

As shown in Figure 1b, we select six typical locations (shown as the points from A to F in Figure 1b in simulating street canyon in CFD to represent the typical positions in the street with different aerodynamically characteristics affected by the arrangement of building layouts. The wind vertical profiles over these positions are also considered to be different from each other. The simulated street canyon in CFD is divided into three sections along the running direction: entrance, middle, and exit. In each section, one point is selected in each side of the street buildings to represent the windward and leeward wind component of three dimension. The vertical wind profiles of wind speeds components ($u$, $v$) at these six typical locations are parameterized and fitted into parameterization equations and can be categorized by different AR and inflow directions.

In each simulating scenario, the wind vertical profiles over the six typical locations were abstracted from the CFD simulations and fitted by empirical functions. The aspect ratio (AR) of street canyon, wind velocity ($v$), and a series of land surface parameters are the key factors of parameterization expressions. Finally, the cluster of wind vertical profiles on the modelling locations A-F were employed in IWSUS scheme. The IWSUS is developed to improve the simulation accuracy of wind vertical profiles in and over the urban canopy by well considering the spatial inhomogeneous of urban canopy.

## 2.3 The definition of the key variables in model

### 2.3.1 The wind vertical profile ( $U_{top}$ and $U_{can}$) in exponential-logarithmic law

The exp-log law wind profile is widely used in wind field parameterization simulation for open and flat natural underlying surface scenarios, which is considered horizontally uniform. The exp-log wind profile is applied in scenarios of urban street canyon, where the logarithmic and exponential part are bounded by the displacement height, which are set as two thirds of buildings' average heights.(Masson, 2000) In this wind profile scheme, the wind velocity at canyon top ($U_{top}$) and the one in the urban canyon ($U_{can}$) are given by Eq. (5) and Eq. (6),

$$U_{top} = \frac{2}{\pi} \frac{\ln\left(\frac{\frac{H}{3}}{z_{0\text{town}}}\right)}{\ln\left(\frac{\Delta z + \frac{H}{3}}{z_{0\text{town}}}\right)} |U_a| \tag{5}$$

$$U_{can} = U_{top} \exp(-N/2), N = 0.5AR \tag{6}$$

where H is the average height of building rooftop, $z_{0\text{town}}$ is dynamic roughness length for canyon system, $\Delta z$ is the height from building roofs to the first atmospheric model level, $U_a$ is the wind velocity at the first level of atmospheric model and $AR$ is aspect ratio.

In this paper, the wind speed profiles by IWSUS scheme are compared with those by the exponential-logarithmic law (exp-log law) in this section.

### 2.3.2 The surface turbulence resistance

The surface turbulence resistance (RES) is the key important variable in the urban canopy model, which mainly affect the surface energy flux and near ground meteorological processes. In IWSUS the RES is calculated based on the vertical profile of wind speed, this method is also widely used in most of the current land surface models. As shown in Eq. (7), the RES by empirical formular in Masson (2000) is expressed as:

$$\text{RES} = \left(11.8 + 4.2\sqrt{U_{can}^2 + W_{can}^2}\right)^{-1} \tag{7}$$

where $U_{can}$ and $W_{can}$ are the horizontal and vertical wind speeds within street canyon. To calculate the flux from overall urban canopy, the Eq. (8) and Eq. (9) are applied first to integrate the overall RES within the urban grid in land surface model.

$$\text{RES}_{pos} = \frac{1}{H} \int \text{RES}_{surf}(z) dz \tag{8}$$

$$\text{RES}_{mean} = \overline{\sum \text{RES}_{pos}}, \tag{9}$$

where subscripts surf stands for the surface of walls, impervious land and building roof, H is building heights. Considering the different wind conditions, the weighted-sum for two orthogonal inlet conditions, perpendicular and parallel to street orientation which are set as the base, is applied for RES estimation.

### 2.3.3 The sensible heat flux over the surfaces

Furthermore, the daily variation of sensible heat flux is also the key index to evaluate the performance of the urban canopy model. The sensible heat fluxes can be calculated based on the RES above and the temperature over each surface in urban canopy, e.g., paved and unpaved land surface, surface of building roof, wall in sun side and shaded side, separately. The simulated sensible heat fluxes both in IWSUS and other urban models are calculated following Eq. (10) to Eq. (12) (Masson, 2000):

$$H_{\text{wall/road}} = C_{p_d}\rho\left(T_{\text{wall/road}} - T_{\text{can}}\right)/\text{RES}_{\text{wall/road}} \tag{10}$$

$$H_{\text{roof}} = C_{p_d}\rho(T_{\text{roof}} - T_{\text{amb}})/\text{RES}_{\text{roof}} \tag{11}$$

$$H = H_{\text{wall}} + H_{\text{road}} + H_{\text{roof}} \tag{12}$$

where $C_{p_d}$ is heat capacity of dry air, $\rho$ is air density and *amb* refers to the air upper than building roofs.

### 2.4 Observation Dataset

A site observation was set up and the data of wind speed were collected and applied in this study: the tower samples of vertical wind profile carried out in the urban canyon and above the building roofs of Nanjing, China during December 22-25, 2017. (Liu et al., 2020; Liu et al., 2021). The location and surrounding environment of the site is shown in Figure 2. It shown in Figure 1 that the experimental observation site is located on a typical urban built-up underlying surface with buildings' orientation mainly from northeast to southwest. The distances between the observation site and the building communities on the north, west and south sides are obviously different, then the west sides are relatively open. Therefore, under different wind direction scenarios, the urban canyon structures have different effects on the wind speed at the observation site. Therefore, we distinguish the wind field heterogeneity in street canyons according to the wind profiles observed under different wind direction scenarios, and compare them with the simulation results in IWSUS.

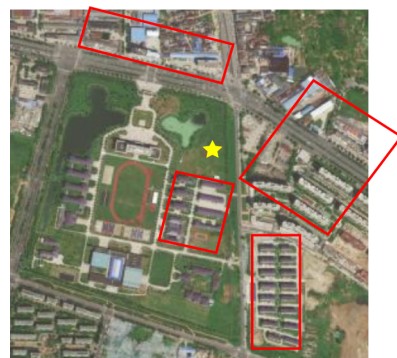

**Figure 2 The location and surrounding environment of the observation site in Nanjing urban area. The yellow star is the location where the vertical wind profiles were measured, and the red rectangles refer to the main buildings surrounding (from Baidu Maps).**

The observation from the boundary-layer wind tunnel experiments by Meteorological Institute at the University of Hamburg, was also applied for validating the wind speed in the street canopy in scenarios of the geometry characters and

orientations of street canyon (Hertwig et al., 2012). The urban geometry in the wind tunnel is set in 1:225-scale model of a semi-idealized urban block with AR ranging from 0.625 to 1.33, which is heterogeneous and morphologically consistent with typical central city characteristics. Due to the similar morphological structure with our CFD model, we apply the vertical
profiles of horizontal wind components at plaza edge and lateral street canyon as the observation at the lower levels of urban canopy for wind speed comparison. Meanwhile, to assess the performance of IWSUS in urban energy balance, the measurement of fluxes over the building canopy and the air temperatures diurnal variation in the street canyon were obtained from the observation research in a deep street canyon (AR=2.1) with both sides approximately equal height buildings in central Gothenburg, Sweden (57°42'N, 11°58'E) by Offerle et al (Offerle et al., 2006). The orientation of the street was approximately
North-South. The measurement instruments were installed at both walls, rooftop and centre respectively. The temperature and wind speed were collected from June 2004 to August 2004 and hourly averaged as the daily variation for east wall, west wall, roof, road and the ambient in summer. The heat flux is calculated based on these observations with Eq. (10) to Eq. (12). We also set the same parameters of the measurement site in IWSUS to obtain the representative simulation results for comparing surface flux under the real situation in section 4.2.

**2.5 Method for Validation**

For validating the simulation accuracy of IWSUS by using the observations mentioned in 2.4, the exponential-logarithm wind profile (referred to as exp-log law in following) scheme widely applied in most of the current urban canopy models (e.g., UCM, Kusaka, 2001) was used as the control group. The index of normalized mean error (NME) was applied for simulated wind speed by both IWSUS and exp-log law to compare with the observations. The NME is defined as follows:

$$NME = \frac{\sum_{i=1}^{N}|C_{sim,i} - C_{obs,i}|}{\sum_{i=1}^{N} C_{obs,i}} \times 100\% \tag{13}$$

where the $C_{sim}$ and $C_{obs}$ represent the values on the spatial grid in simulation (either IWSUS or exp-log law) and observation, respectively. The subscript $i$ indicates the numbers of vertical layer in the model grids.

The simulation results of turbulence heat flux in and over the urban canopy based on the UCM and IWSUS were also validated by the NME index.

**3 Simulation Results**

**3.1 CFD simulation for Inflow scenarios**

In each group of scenarios cataloged in Table.1, the 3-Dimension wind field was calculated in the CFD method based on different initial parameters in Table 1. Then the vertical wind profiles of the horizontal components ($u$, $v$) of the wind speed over the six representative positions were parameterized, respectively. It is noted that the simulation results of 1 m/s and 8 m/s
scenarios are very close after normalized by inflow speed, so the following figures only shows the inflow scenarios of 0.1 m/s

and 8 m/s. The original and processed results of wind profiles at different positions can be found in Zenodo links in Code and data availability section.

As shown in Figure 3 and Figure 4, the absolute value of $u$ and $v$ represents the wind speed, and the positive and negative signs indicate that the wind direction at this height is the same as or opposite to the wind speed in the free atmosphere above the building canopy.

The vertical profiles of $u$ and $v$ component by CFD simulations at each representative point when the inflow is perpendicular to the street direction are shown in Figure 3. First, the vertical profiles of the $u$ component at the six representative points generally show similar structure: in the height range below the building height (H), the $u$ value is small and the value varies little with height; while around the height of H, the u value increases dramatically. It indicates that the u component of the wind speed has an obvious wind shear effect near the average height of the building canopy. Besides, the direction of $u$ also reverses around the height of H: above H, the $u$ component is in the same direction as the upper free atmosphere, while below is opposite.

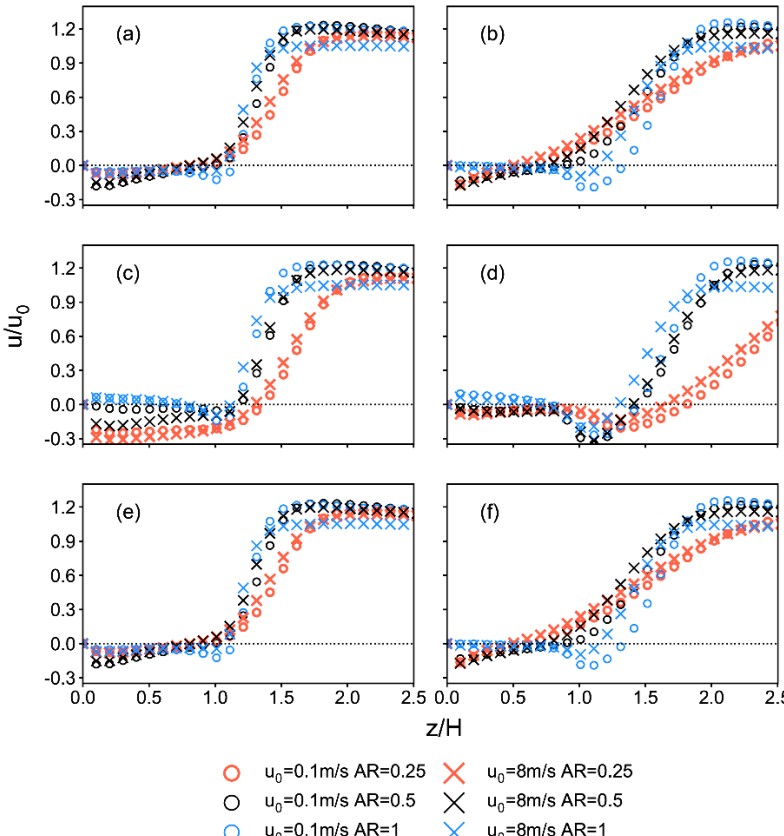

Figure 3 Simulating wind vertical profiles of $u$ component for perpendicular scenarios at six sampling points. Circle, and cross (x) represents the inflow of 0.1 m/s and 8 m/s respectively. Red, black, and blue scatters represent AR=0.25, 0.5 and 1. Subplot (a)-(f) represent six sample positions A-F sampled in Figure 1b respectively. The sign of $u/u_0$ represents direction along $x$ axis shown in Figure 2.

The vertical profiles of *u* values over different representative points are somewhat different in details: the vertical profiles of *u* at the windward side (Figure 3b, d, f) change more dramatically near the height H than those at the leeward side (Figure

3a, c, e), and the increasing above the height H on the windward side is slower than that on the leeward side under the same AR. Due to the spatial symmetry setting of the street canyon shape in CFD, the simulated *u* values at the entrance and exit of the street canyon are quite similar.

The values of *v* component at both ends of street in the leeward side generally increase first reaching their maximum near the height of H, and then gradually decrease with the increase of height tending to near 0 at the position far above the top of

the building (Figure 4a, e). At both ends of the windward side of the street canyon, because the horizontal airflow is stronger after bypassing the building, the *v* component has a larger value near the ground when AR is small, and maintains a similar change as that at the leeward side when AR is large (Figure 4b, f). The wind almost flows along the street canyon due to the obstruction of the building facade (Figure 4c, d). Similarly, due to the geometric symmetry of the model, the *v* components at both ends of the street valley are similar in value but opposite in direction.

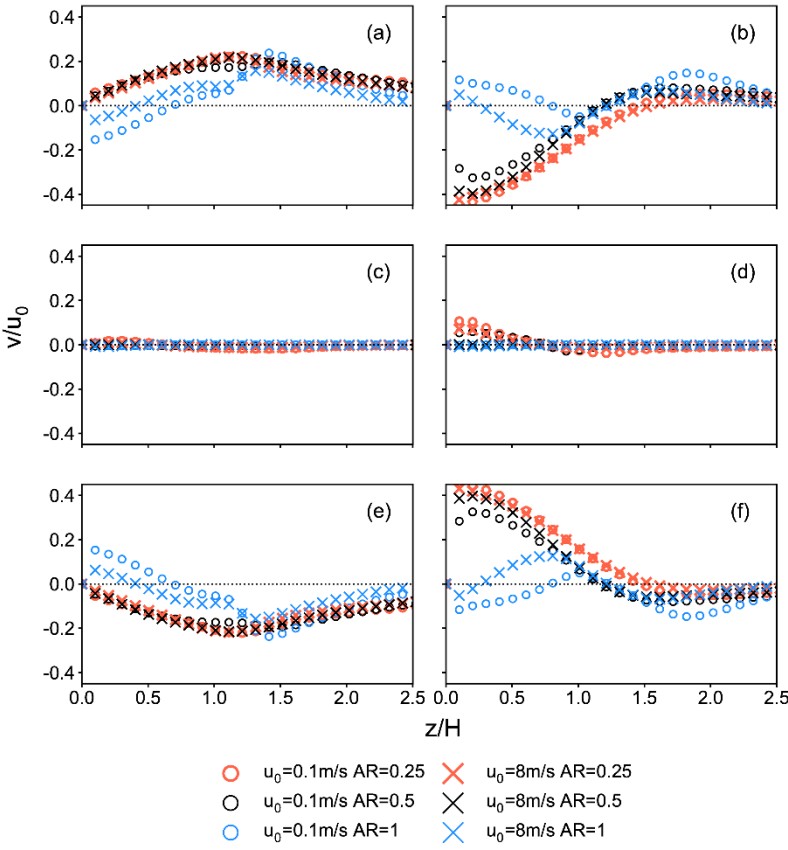

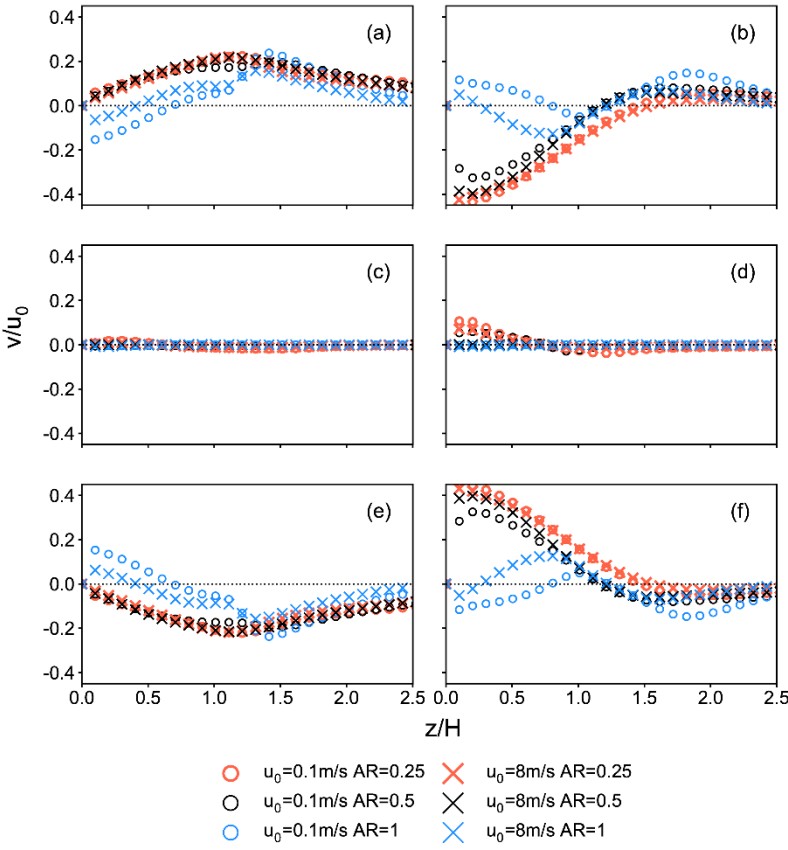


**Figure 4 Simulating wind vertical profiles of *v* component for perpendicular scenarios at six sampling points A-F. The representations of shapes and colors are same as Figure 3. The sign of *v*/$u_0$ represents direction along *y* axis shown in Figure 1.**

In the scenario group 1 that the inflow is perpendicular to the street canyon, the CFD simulation results generally show that the horizontal wind speed is low and changes gently below the height of H. The wind speed then increases rapidly near the height of H forming a strong vertical shear of the horizontal wind speed, and finally approaches the background wind speed when the height is far above H. The simulation results by CFD indicate that the vertical profiles of horizontal wind speed in and above the street canyon mostly do not conform to the traditional logarithmic-exponential wind profiles. The vertical profiles of wind speed over the different positions in street canyons are close to each other but quite different in details under different distributions of surrounding buildings. The results support the view that there is heterogeneous distribution of wind field in and over the inhomogeneous street canyon. Therefore, the vertical wind profile scheme, with exponential in lower layer and logarithmic in upper-layer, which is widely used in land surface models cannot well simulate the wind spatial distributions in and over the urban canopy.

The vertical profiles for $v$ component over the six representative positions by CFD simulations in scenario group 2, in which the inflow is set to be parallel to street orientation, are shown in Figure 5. The results of $u$ component show convergence in the entrance because of waking flows from building blocking while relatively small values at the middle and the exit. Due to the lack of blocking effects to $v$ component in parallel scenarios, the value of $v$ component is higher and increase rapidly with height at lower levels in street canyon, which is obvious at the entrance. As reaching to the half of building height, the increasing rate is much smaller and gradually approach to the inflow speed. The shapes of these profiles are not only different from those in results of scenarios group 1, but also do not accord with the law of Exponential-Logarithmic in most of current urban canopy schemes. The height at which vertical wind shear occurs above each of the six representative points also show obviously differences. Thus, the heterogeneity and complexity of the horizontal and vertical distribution of the wind field in the urban building-street canopy are well reflected in these CFD simulations, which has been proved in series of observations (Castro, 2017; Liu et al., 2020; Liu et al., 2021).

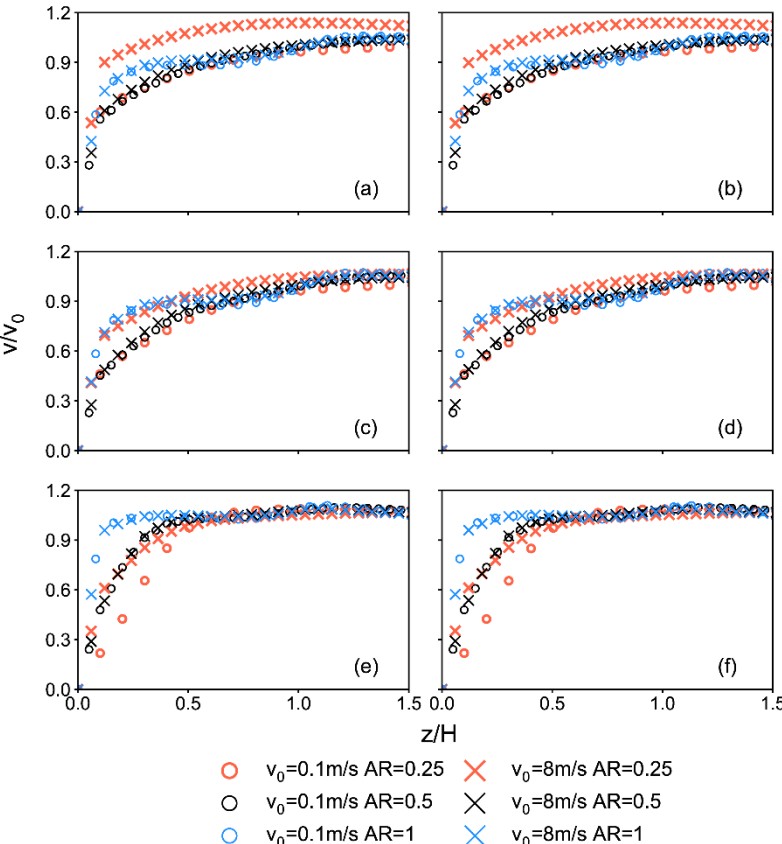

Figure 5 Simulating wind vertical profiles of $v$ component for parallel scenarios at six sampling points A-F. The representations of shapes and colors are same as Figure 3. The sign of $v/v_0$ represents direction along $y$ axis shown in Figure 1.

## 3.2 Piecewise fitting for wind profile

We employed the piecewise function to fit the vertical profiles of horizontal wind obtained from CFD simulation. The aspect ratio ($AR$) of street canyon is a key important parameter to qualify the degree of urbanization in the urban canopy model. As shown in Figure 3 and Figure 4, when the inflow wind speed keeps the same, the vertical profiles of the $u$ and $v$ components vary with $AR$: when the value of $AR$ increases, $u$ decreases below the height of H and increases near the H. The $v$ component averagely has a larger value when $AR$ is smaller. Finally, we get the normalized regression expression between the wind speed ($u_r = u/u_0$, $v_r = v/u_0$), relative vertical height ($h_r$), and $AR$. The $h_r$ defines as the height ($z$) compared with the height of buildings' top (H):

$$h_r = z/H \tag{14}$$

where H represents the average height of buildings' roof in the urban model. The parametric fit to the vertical profile of the wind component $u$ is piecewise functions combined by four segments for each represented points in street canyon, which are near ground section, inner street canyon section, the building roof section, and the above the roof section, divided by three

segmentation points (Marked as $H_1$, $H_2$, $H_3$ and etc.). After determine the normalized height ($h_r$) and normalized wind
component ($u_r$ etc.) at each segmentation points, the type of function for every piece at different heterogeneous representative
positions are specified according to the variation style. The coefficients in these specified functions are solved by substituting
the obtained height and wind component values at segmentation points in both ends for every piece. Finally, the expressions
of wind profiles are obtained. Here, the fitting results of horizontal components ($u$ and $v$) in perpendicular scenarios are shown
in this section as an example. The other scenarios are listed in the supplements.

Table 2 and Table 3 list the expressions of normalized height ($h_r$) and normalized $u$ component ($u_r$) at each segmentation
points respectively for different typical locations in perpendicular scenario. The expressions of $u$ component vertical profile
for the perpendicular scenarios are listed in Table 4. Figure 6 shows the vertical profiles of the $u$ component over the six
representative points obtained by fitting methods. The curves in different colors represent the fitting results with different $AR$
parameters listed in the Table 1 respectively. The results show that the values of $u$ component are relatively small in the vertical
range from ground to around the height of H. It illustrates that the buildings have strong dragging effect on the air flow
perpendicular to the street orientations in the urban street canyons, which results in lower wind speed of near the ground. The
value and sign of $u$ component varies rapidly near the height of H, which illustrates of strong wind shear pattern near the
average height of buildings' roof. In addition, the larger the value of $AR$, the more intense the wind shear, indicating that there
is a positive correlation between the development of urban and the complexity of wind field in the urban street canyon.


**Table 2 The fitting functions of the segmentation points for the wind components *u* in the perpendicular scenarios**

| Positions | Segmentation points | Fitting Functions |
|---|---|---|
| leeward at entrance/exit | $H_1$ | $h_r = 0.2781 \exp(-0.8123AR)$ |
| | $H_2$ | $h_r = 1$ |
| | $H_3$ | $h_r = 1.712 \exp(-5.94AR) + 1.612$ |
| windward at entrance/exit | $H_1$ | $h_r = 0.1$ |
| | $H_2$ | $h_r = \begin{cases} -1.278 \exp(-2.643AR) + 0.8242, AR \geq 0.5 \\ -0.8667AR + 0.9167, AR < 0.5 \end{cases}$ |
| | $H_3$ | $h_r = 0.4612 \exp(-4.901AR) + 1.364$ |
| leeward at middle | $H_1$ | $h_r = \begin{cases} -0.019 \exp(2.563AR) + 0.304, AR \leq 1 \\ h_r(AR = 1), AR > 1 \end{cases}$ |
| | $H_2$ | $h_r = 1$ |
| | $H_3$ | $h_r = 1.069 \exp(-2.746AR) + 1.254$ |
| windward at middle | $H_1$ | $h_r = \begin{cases} 0.467AR + 0.083, AR \leq 0.5 \\ 4.551 \exp(-5.55AR) + 0.032, AR > 0.5 \end{cases}$ |
| | $H_2$ | $h_r = 0.8$ |
| | $H_3$ | $h_r = 3.29 \exp(-5.56AR) + 1.512$ |

**Table 3 The fitting functions of the wind components *u* at the segmentation points ($H_i$) in the perpendicular scenarios**

| Positions | Segmentation Points | Fitting Functions |
|---|---|---|
| leeward at entrance/exit | $H_1$ | $u_r = \begin{cases} -0.538 \exp(0.5AR) + 0.53, AR \leq 0.5 \\ 0.194AR - 0.2586, 0.5 < AR < 1 \\ 0.093 \exp(-1.087AR) - 0.096, AR \geq 1 \end{cases}$ |
| | $H_2$ | $u_r = 0.289 \exp(-1.124AR) - 0.1473$ |
| | $H_3$ | $u_r = 1.15$ |
| windward at entrance/exit | $H_1$ | $u_r = 0.3515 \exp(-0.908AR) + 0.099$ |
| | $H_2$ | $u_r = \begin{cases} -0.493AR + 0.185, AR < 0.5 \\ -0.354 \exp(-0.245AR) + 0.2514, 0.5 \leq AR \leq 1 \\ u_r(AR = 1), AR > 1 \end{cases}$ |
| | $H_3$ | $u_r = 0.569$ |
| leeward at middle | $H_1$ | $u_r = -0.649 \exp(-2.17AR) + 0.107$ |
| | $H_2$ | $u_r = \begin{cases} -0.495AR - 0.327, AR < 0.5 \\ 0.259 \exp(-3.461AR) - 0.119, AR \geq 0.5 \end{cases}$ |
| | $H_3$ | $u_r = \begin{cases} 0.118AR + 0.878, AR < 0.5 \\ 0.412 \exp(-1.28AR) + 0.719, AR \geq 0.5 \end{cases}$ |
| | $H_1$ | $u_r = \begin{cases} 0.0134 \exp(2.66AR) - 0.109, AR \leq 1 \\ u_r(AR = 1), AR > 1 \end{cases}$ |

| | | |
|---|---|---|
| windward at middle | $H_2$ | $u_r = \begin{cases} -0.045\text{AR} - 0.029, \text{AR} < 0.5 \\ -0.244\exp(-3.83\text{AR}) - 0.015, \text{AR} \geq 0.5 \end{cases}$ |
| | $H_3$ | $u_r = 0.604$ |

**Table 4 Expressions of $u$ component vertical profile for perpendicular scenarios, where the functional relations between $u_r$ and $h_r$ and determine functions of coefficients in these expressions are listed. The $H_i$ and $u_r(H_i)$ represents the relative height and relative wind components speed at the endpoints determined by Table 2 and Table 3 respectively.**

| Positions | Expressions of wind profile | Determine functions of coefficients in expressions |
|---|---|---|
| leeward at entrance/exit | $u_r(h_r)$ $= \begin{cases} \dfrac{a_{1,1}}{\ln(h_{r+1})}, 0 \leq h_r \leq H_1 \\ a_{1,2}(h_r + b_{1,2})^2 + c_{1,2}, H_1 \leq h_r \leq H_2 \\ a_{1,3}(h_r + b_{1,3})^2 + c_{1,3}, H_2 \leq h_r \leq H_3 \\ \dfrac{a_{1,4}\exp(b_{1,4}h_r) + u_0}{u_0}, h_r \geq H_3 \end{cases}$ | $\begin{cases} a_{1,1} = \dfrac{u_r(H_1)}{ln(h_r(H_1) + 1)} \\ a_{1,2} = \dfrac{u_r(H_2) - c_{1,2}}{(h_r(H_2) + b_{1,2})^2}, b_{1,2} = -h_r(H_1), c_{1,2} = u_r(H_1) \\ a_{1,3} = \dfrac{u_r(H_2) - c_{1,3}}{(h_r(H_2) + b_{1,3})^2}, b_{1,3} = -h_r(H_3), c_{1,3} = u_r(H_3) \\ a_{1,4} = \dfrac{u_r(H_3) - u_0}{exp(b_{1,4}h_r(H_3))}, b_{1,4} = -0.1 \end{cases}$ |
| windward at entrance/exit | $u_r(h_r)$ $= \begin{cases} \dfrac{a_{2,1}}{\ln(h_{r+1})}, 0 \leq h_r \leq H_1 \\ a_{2,2}h_r + b_{2,2}, H_1 \leq h_r \leq H_2 \\ a_{2,3}(h_r + b_{2,3})^2 + c_{2,3}, H_2 \leq h_r \leq H_3 \\ \dfrac{a_{2,4}\exp(b_{2,4}h_r) + u_0}{u_0}, h_r \geq H_3 \end{cases}$ | $\begin{cases} a_{2,1} = \dfrac{u_r(H_1)}{ln(h_r(H_1) + 1)} \\ a_{2,2} = \dfrac{u_r(H_2) - u_r(H_1)}{h_r(H_2) - h_r(H_1)}, b_{2,2} = u_r(H_2) - a_{2,2}h_r(H_2) \\ a_{2,3} = \dfrac{u_r(H_3) - c_{2,3}}{(h_r(H_3) + b_{2,3})^2}, b_{2,3} = -h_r(H_2), c_{2,3} = u_r(H_2) \\ a_{2,4} = \dfrac{u_r(H_3) - u_0}{exp(b_{2,4}h_r(H_3))}, b_{2,4} = -0.1 \end{cases}$ |
| leeward at middle | $u_r(h_r)$ $= \begin{cases} \dfrac{a_{3,1}}{\ln(h_{r+1})}, 0 \leq h_r \leq H_1 \\ a_{3,2}(h_r + b_{3,2})^2 + c_{3,2}, H_1 \leq h_r \leq H_2 \\ a_{3,3}(h_r + b_{3,3})^2 + c_{3,3}, H_2 \leq h_r \leq H_3 \\ \dfrac{a_{3,4}\exp(b_{3,4}h_r) + u_0}{u_0}, h_r \geq H_3 \end{cases}$ | $\begin{cases} a_{3,1} = \dfrac{u_r(H_1)}{ln(h_r(H_1) + 1)} \\ a_{3,2} = \dfrac{u_r(H_2) - c_{3,2}}{(h_r(H_2) + b_{3,2})^2}, b_{3,2} = -h_r(H_1), c_{3,2} = u_r(H_1) \\ a_{3,3} = \dfrac{u_r(H_3) - c_{3,3}}{(h_r(H_3) + b_{3,3})^2}, b_{3,3} = -h_r(H_2), c_{3,3} = u_r(H_2) \\ a_{3,4} = \dfrac{u_r(H_3) - u_0}{exp(b_{3,4}h_r(H_3))}, b_{3,4} = -0.2 \end{cases}$ |
| windward at middle | $u_r(h_r)$ $= \begin{cases} \dfrac{a_{4,1}}{\ln(h_{r+1})}, 0 \leq h_r \leq H_1 \\ a_{4,2}(h_r + b_{4,2})^2 + c_{4,2}, H_1 \leq h_r \leq H_2 \\ a_{4,3}(h_r + b_{4,3})^2 + c_{4,3}, H_2 \leq h_r \leq H_3 \\ \dfrac{a_{4,4}\exp(b_{4,4}h_r) + u_0}{u_0}, h_r \geq H_3 \end{cases}$ | $\begin{cases} a_{4,1} = \dfrac{u_r(H_1)}{ln(h_r(H_1) + 1)} \\ a_{4,2} = \dfrac{u_r(H_2) - c_{4,2}}{(h_r(H_2) + b_{4,2})^2}, b_{4,2} = -h_r(H_1), c_{4,2} = u_r(H_1) \\ a_{4,3} = \dfrac{u_r(H_3) - c_{4,3}}{(h_r(H_3) + b_{4,3})^2}, b_{3,3} = -h_r(H_2), c_{3,3} = u_r(H_2) \\ a_{4,4} = \dfrac{u_r(H_3) - u_0}{exp(b_{4,4}h_r(H_3))}, b_{4,4} = -0.3 \end{cases}$ |

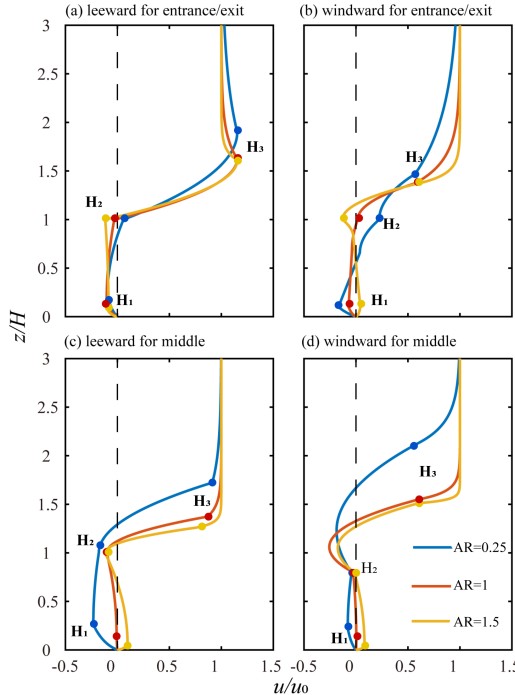

**Figure 6 Wind profile of *u* component obtained from IWSUS at different positions with AR= 0.25, 1 and 1.5. The point Hᵢ are the segmentation points referred in Table 2 and Table 3.**

Table 5 and Table 6 list the fitting formulas for the *v* component of the wind speed on the leeward side of the building based on CFD simulation. The fitted relative height ($h_r$) of the segment points of the fitting curve are calculated by the formulas in Table 5, and the specified equations used to describe the vertical wind profiles $v_r$ between the segment points of $h_r$ are listed in Table 6. Then the piecewise fitting method similar as *u* component mentioned above is also applied for *v* component profiles.

**Table 5 The fitting functions of the segmentation points for the wind components *v* in the perpendicular scenarios**

| Positions | Segmentation Points | Fitting Functions |
|---|---|---|
| leeward at entrance/exit | $H_1$ | $h_r = 1$ |
| windward at entrance/exit | $H_1$ | $h_r = \begin{cases} h_r(\text{AR} = 0.5), \text{AR} < 0.5 \\ 1.52\exp(-4.39\text{AR}) + 0.0313, \text{AR} \geq 0.5 \end{cases}$ |
| | $H_2$ | $h_r = \begin{cases} -2.55\exp(0.261\text{AR}) + 3.722, \text{AR} \leq 0.5 \\ h_r(\text{AR} = 0.5), \text{AR} > 0.5 \end{cases}$ |
| | $H_3$ | $h_r = \begin{cases} -2.39\exp(0.545\text{AR}) + 4.673, \text{AR} \leq 0.5 \\ h_r(\text{AR} = 0.5), \text{AR} > 0.5 \end{cases}$ |

**Table 6 The fitting functions of the segmentation points (H$_i$) for the wind components $v$ in perpendicular scenarios**

| Positions | Segmentation Points | Fitting Functions |
|---|---|---|
| leeward at entrance | H1 | $v_r = -0.07\exp(-1.58\text{AR}) - 0.173$ |
| windward at entrance | $H_1$ | $v_r = \begin{cases} -0.03\exp(2.98\text{AR}) + 0.503, \text{AR} \le 1 \\ v_r(\text{AR}=1), \text{AR} > 1 \end{cases}$ |
| | $H_2$ | $v_r = 0.15$ |
| | $H_3$ | $v_r = 0.1326\exp(-2.77\text{AR}) - 0.101$ |
| leeward at exit | H1 | $v_r = 0.07\exp(-1.58\text{AR}) - 0.173$ |
| windward at exit | $H_1$ | $v_r = \begin{cases} 0.03\exp(2.98\text{AR}) + 0.503, \text{AR} \le 1 \\ v_r(\text{AR}=1), \text{AR} > 1 \end{cases}$ |
| | $H_2$ | $v_r = -0.15$ |
| | $H_3$ | $v_r = -0.1326\exp(-2.77\text{AR}) - 0.101$ |

**Table 7 Expressions of $v$ component vertical profile for perpendicular scenarios, where the functional relations between v$_r$ and h$_r$ and determine functions of coefficients in these expressions are listed. The H$_i$ and v$_r$(H$_i$) represents the relative height and relative wind components speed at the endpoints determined by Table 5 and Table 6 respectively.**

| Positions | Expressions of wind profiles | Determine functions of coefficients in expressions |
|---|---|---|
| leeward at entrance | $v_r(h_r)$ $= \begin{cases} \dfrac{a_{1,1}}{\ln(h_r+1)}, 0 \le h_r \le H_1 \\ a_{1,2}(h_r + b_{1,2})^2 + c_{1,2}, H_1 \le h_r \le H_2 \\ \dfrac{a_{1,3}\exp(b_{1,3}h_r)}{u_0}, h_r \ge H_2 \end{cases}$ | $\begin{cases} a_{1,1} = \dfrac{v_r(H_1)}{ln(h_r(H_1)+1)} \\ a_{1,2} = \dfrac{v_r(H_1) - c_{1,2}}{(h_r(H_1)+b_{1,2})^2}, b_{1,2} = -h_r(H_2), c_{1,2} = u_r(H_2) \\ a_{1,3} = \dfrac{v_r(H_2)}{exp(b_{1,3}h_r(H_2))}, b_{1,3} = -0.1 \end{cases}$ |
| windward at entrance | $v_r(h_r)$ $= \begin{cases} \dfrac{a_{2,1}}{\ln(h_r+1)}, 0 \le h_r \le H_1 \\ a_{2,2}(h_r + b_{2,2})^2 + c_{2,2}, H_1 \le h_r \le H_2 \\ \dfrac{a_{2,3}\exp(b_{2,3}h_r)}{u_0}, h_r \ge H_2 \end{cases}$ | $\begin{cases} a_{2,1} = \dfrac{v_r(H_1)}{ln(h_r(H_1)+1)} \\ a_{1,2} = \dfrac{v_r(H_2) - c_{1,2}}{(h_r(H_2)+b_{1,2})^2}, b_{1,2} = -h_r(H_1), c_{1,2} = u_r(H_1) \\ a_{1,3} = \dfrac{v_r(H_2) - c_{1,3}}{(h_r(H_2)+b_{1,3})^2}, b_{1,3} = -h_r(H_3), c_{1,2} = u_r(H_3) \\ a_{1,4} = \dfrac{v_r(H_3)}{exp(b_{1,4}h_r(H_3))}, b_{1,4} = -0.1 \end{cases}$ |
| leeward at exit | $v_r(h_r)$ $= \begin{cases} \dfrac{a_{1,1}}{\ln(h_r+1)}, 0 \le h_r \le H_1 \\ a_{1,2}(h_r + b_{1,2})^2 + c_{1,2}, H_1 \le h_r \le H_2 \\ \dfrac{a_{1,3}\exp(b_{1,3}h_r)}{u_0}, h_r \ge H_2 \end{cases}$ | $\begin{cases} a_{1,1} = \dfrac{v_r(H_1)}{ln(h_r(H_1)+1)} \\ a_{1,2} = \dfrac{v_r(H_1) - c_{1,2}}{(h_r(H_1)+b_{1,2})^2}, b_{1,2} = -h_r(H_2), c_{1,2} = u_r(H_2) \\ a_{1,3} = \dfrac{v_r(H_2)}{exp(b_{1,3}h_r(H_2))}, b_{1,3} = -0.1 \end{cases}$ |

$$\text{windward at exit} = v_r(h_r) \begin{cases} \dfrac{a_{2,1}}{\ln(h_r+1)}, 0 \le h_r \le H_1 \\ a_{2,2}(h_r + b_{2,2})^2 + c_{2,2}, H_1 \le h_r \le H_2 \\ \dfrac{a_{2,3}\exp(b_{2,3}h_r)}{u_0}, h_r \ge H_2 \end{cases} \qquad \begin{cases} a_{2,1} = \dfrac{v_r(H_1)}{ln(h_r(H_1)+1)} \\ a_{1,2} = \dfrac{v_r(H_2) - c_{1,2}}{(h_r(H_2) + b_{1,2})^2}, b_{1,2} = -h_r(H_1), c_{1,2} = v_r(H_1) \\ a_{1,3} = \dfrac{v_r(H_2) - c_{1,3}}{(h_r(H_2) + b_{1,3})^2}, b_{1,3} = -h_r(H_3), c_{1,2} = v_r(H_3) \\ a_{1,4} = \dfrac{v_r(H_3)}{exp(b_{1,4}h_r(H_3))}, b_{1,4} = -0.1 \end{cases}$$

The expressions of $u$ component vertical profile for the perpendicular scenarios are listed in Table 7, and Figure 7 shows the fitting curves of $v$ component of wind speed based on IWSUS simulations. The settings of $AR$ are the same as Figure 6. On the one hand, the simulated wind speed over different representative points varies largely in both the size of value and the direction, indicating the heterogeneity of wind speed field in the street valley. On the other hand, the $AR$ is a sensitive parameter,

significantly affecting the shape of the $v_r$-$h_r$ curve. It indicates that the development of urban building-street canopy has an impact on the vertical distribution of wind speed.

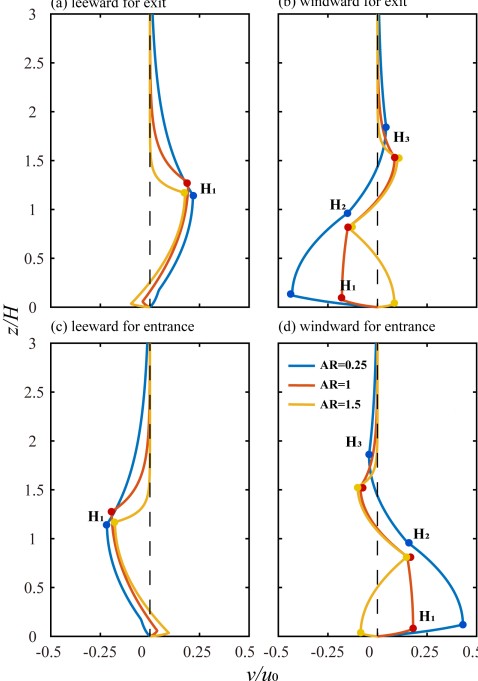

**Figure 7 Wind profile of $v$ component obtained from IWSUS at different positions with AR= 0.25, 1 and 1.5. The point Hᵢ are the segmentation points referred in Table 2 and Table 3.**

The $u$ and $v$ components of wind speed simulated by IWSUS are very sensitive to the width of the street. Inside a narrow street, the wind is relative weaker, which refers to the stronger blocking effects. The wind direction is more variable at different levels, indicating different exchange process between building surface and the air inside street. In the middle part, the direction at lower levels is same as inflow, capable of leading to heat transfer from leeward to windward; while the opposite direction for higher levels near building roof refers a reversed heat transfer. Due to the velocity is larger at upper part of the canyon, the

heat transfer mode can cause the warmer air in the top and cooler in the near-ground with the same heat emission from both leeward and windward, which refers the trend of invasion in urban canopy and is unfavourable for vertical pollution diffusion. In the ends, the wind tends to flow out of the street at lower levels, which infers that the mass and energy transport outside of urban area horizontally at ground level.

    For the scenario of wider street, the wind is relatively stronger for the weaker blocking effects from the sparser building
distribution. The direction of u inside street is almost from windward to leeward, and the scale in leeward is larger, which is favourable of accumulating heat and mass in lee side, leading to the heterogeneous distribution of meteorological fields. In both ends, the inflow at windward is stronger than the outflow at leeward, also promoting the heat and mass convergence in urban area.

    In general, the wind condition based on IWSUS is in favour of gathering heat and energy in urban area and the weaker
wind and probably existence of invasion layer in narrow street might be more capable of accumulating heat and air pollutants. For the scenarios of inflow parallel to street orientation, the specific fitting results can be found in the supplements.

## 4 Validation of IWSUS

### 4.1 Comparison of IWSUS and Observation

    Figure 8 illustrates the verification of the inhomogeneous effects by validating the wind vertical profile and validate the
wind speed profiles in IWSUS simulation by the observation. Figure 8a shows the arrangement of buildings around the observation point. As the heterogeneity of urban canopy and wind speed in and over the observation spots are fully considered in the IWSUS, a proper wind speed vertical profile can always be found in IWSUS to match the surrounding building arrangements around the observation position. Specifically for this example, the observation point is closer to the southeast side of the building in the street canyon, we take the simulating wind over the representative position in leeward side and the
windward side in the street canyon in IWSUS to correspond to the southeast wind and northwest wind scenarios in the observation data.

    As shown in Figure 8b and Figure 8c, when the leeward scenario (southeast wind happened, as shown in Figure 8b) triggered, the result by IWSUS is generally more consistent with the observation than that by exp-log law. The horizontal wind speed below the building height maintains a low value as 0.098 in IWSUS, 0.055 in observation, and 0.472 by exp-log law,
indicating that the performance of IWSUS is much better than that of the exp-log law. The NME analysis of wind speed shows that the error between IWSUS and observation is 67.0% from the surface to four times the height of buildings, which is slightly lower than that in exp-log law, 73.1%. For the interior of the street canyon, the improvement of IWSUS is more obvious, from 756.3% by exp-log law to 78.0% by IWSUS, which shows that IWSUS can well grasp the wind profile characteristics of the leeward side in the street canyon.

In the scenario of the windward side (northwest wind, Figure 8c), the trend of IWSUS is basically consistent with the observation. The averaged wind speed by IWSUS is lower than observation in the street canyon while that by exp-log law is

higher. NME analysis shows that the error between IWSUS and observation is 23.4% from the surface to four times the height of the building, which is also slightly better than that by exp-log law, 33.9%. The improvement of the wind field inside the street canyon is obvious, from 196.2% by exp-log law to 45.6% by IWSUS.

To sum up, the wind field simulation results by IWSUS in the street canyon fully consider the inhomogeneous characteristics of the near-surface wind field caused by the building street canyon in the urban canopy. Thus, the wind profile by IWSUS shows better agreement with the observed wind speed in the street canyon, compared with the wind profile by exp-log law.

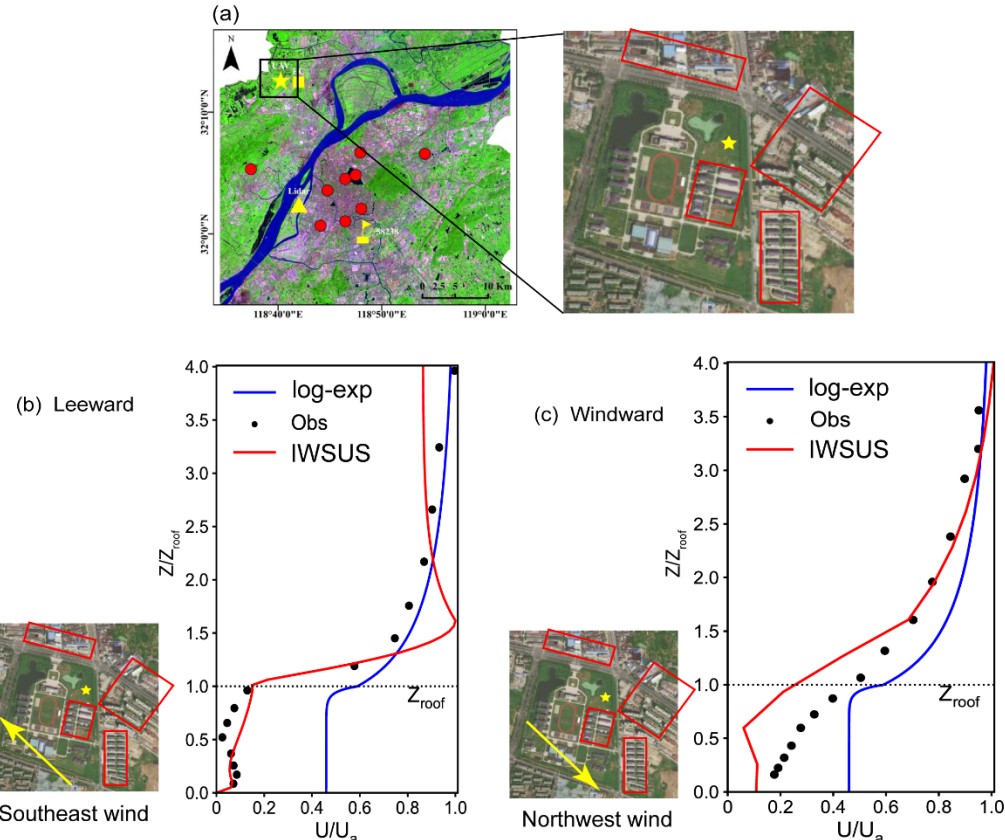

**Figure 8 Overall wind vertical profiles comparison between IWSUS, exp-log law and observations in various wind direction scenarios. The location of the observation point is marked as a yellow pentagram (Liu et al., 2020; Liu et al., 2021) in the left of Figure 8a with the specific surrounding environment (from Baidu Maps) in the right; the red boxes are the main existence of the high-rise buildings nearby; Figure 8b and 8c show the vertical wind profile obtained from IWSUS (red line)m exp-log (blue line) and observations (black dots) at the observation site under leeward side with southeast wind scenarios and windward side with** 385 **northwest wind scenarios, respectively.**

Figure 9 shows the horizontal averaged wind vertical profiles by overall grids of urban in IWSUS, exp-log law and the observations. Below the building roof level, the wind speed from IWSUS is obviously closer to the observation than that by exp-log law, which were obviously overestimated. Thus, the IWSUS show large improvement in the simulation of the averaged wind speed in street canyon. This is mainly because in IWSUS, the effects to the wind speed in street canyon by the

inhomogeneity of urban canopy, e.g., street orientation, the background wind speed, and the differences of typical positions in street canyon as set up in section 2.2, are fully considered. Thus, the various wind speed in different oriented streets, some of them even with the opposite direction at different locations in the street, results in the lower averaged wind speed by IWSUS in the model grid than that by the logarithmic-exponential law. The vertical variation of the wind speed in the street canyon both by the IWSUS and logarithmic-exponential law are relatively small. More deviation of wind speed by IWSUS compared

to observation than that of exp-log law in and around the height of H. It probably relates to the differences of real street canyon and our CFD setting: the buildings are set to be with the same height in our CFD scenarios while the building height somehow varies around the site of observation. For the levels upper than building roof, both IWSUS and exp-log law coincide with the observations, because the inhomogeneity of the atmospheric flow field caused by the street canyon array is weak in the upper layers higher than the average height of buildings.

However, the thermal heterogeneity of the surfaces in the street canyon have not been further considered in our current study. The main purpose to replace the exp-log scheme with the IWSUS is to fully considering the aerodynamical heterogeneity in and over the urban street canyon. But the buoyancy and thermodynamics conditions play also very important roles in the wind and turbulence simulation over the urban areas. It should be focused on in our future study.

         The normalized mean error (NME) is applied in our study to evaluate the wind speed error between simulation and

observation. The NME of wind speed by IWSUS is 49.0%, compared to 56.1% by exp-log law for the total vertical layers in the range from the ground to four times the average height of the building. For the vertical averaged wind speed in the street canyon (range from the ground to building top), the NME of wind speed by IWSUS is 70%, while those by exp-log law is 285.8%. These results prove that the improvement in the vertical wind profiles by IWSUS is obvious compared with the exp-log scheme, which is widely used in most of the current urban model.

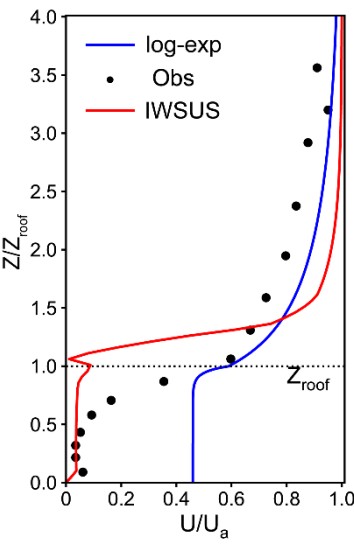


**Figure 9 Overall averaged wind vertical profiles by IWSUS, exp-log law and the observation.**

## 4.2 Comparison of surface flux for daily variation

Several modeling studies have shown that the overestimation of the surface fluxes and the lack of aerodynamical blocking effects from building arrays might be the main reasons for higher wind speed in the bottom of urban street canyons (Salamanca et al., 2018; Zhang et al., 2019). Meanwhile, the overestimated wind can also result in the underestimation of peak levels of air pollutants in air quality models since the stronger winds are favorable to the diffusion of pollutants (Jin et al., 2021; Ulpiani, 2021). Therefore, the performances of simulated surface energy flux by IWSUS are also validated in our study.

Table 8 gives the results of RES estimation by IWSUS and Masson with the aspect ratio equal to 1. Considering the different wind conditions, the weighted-sum for two orthogonal inlet conditions, perpendicular and parallel to street orientation which are set as the base, is applied for RES estimation.

**Table 8 Average resistance (RES) obtained from IWSUS and exp-log for urban surface with AR=1. The RES of wall, road and roofs are calculated from IWSUS.**

| $RES_{wall}$ | $RES_{road}$ | $RES_{roof}$ | $RES_{exp\text{-}log}$ |
|---|---|---|---|
| 0.0782 | 0.0700 | 0.0473 | 0.0343 |

The results show that IWSUS gives larger RES over the different surfaces in the urban canopy than exp-log law does, because the aerodynamic drag effect of the street canyon array of urban buildings are fully considered. Thus, IWSUS describes a lower wind speed effect in the street canyon perpendicular to the background wind field.

Furthermore, the daily variation of sensible heat flux is also calculated in IWSUS. The validation were based on observation in street canyon at central Gothenburg, Sweden (57°42'N, 11°58'E) in summer time (Offerle et al., 2006). Both the hourly temperature and sensible heat flux among that street canyon were obtain from the observation, while the simulated sensible heat fluxes from IWSUS and the exp-log law in Masson's study are calculated with Eq. (10) to Eq. (12).

Figure 10 shows the daily variations of the sensible heat fluxes in IWSUS and by Masson's logarithmic-exponential wind profile scheme, respectively. The observed sensitive fluxes in urban street canyons were also applied to verify. The multiple options of wind directions and representative points were set to coincide with the surrounding environment of observation locations. The red solid line in the figure is the result by IWSUS method, the blue one is based on the logarithmic-exponential scheme, and the black dotted line is the observation. It is indicated that diurnal variations of the sensitive fluxes by IWSUS are obviously more consistent with the observation than those by the log-exp law: it is 32.3% lower in the daytime and 36.3% at night by IWSUS than by the log-exp law, respectively. Accordingly, if the IWSUS scheme is applied in land surface model, the rise rate of near ground temperature over the urban area during the day and the decline rate at night can be partly improved since more rapid rising during the daytime and the dropping at night usually happens in most of the current mainstream land surface model for urban grids.

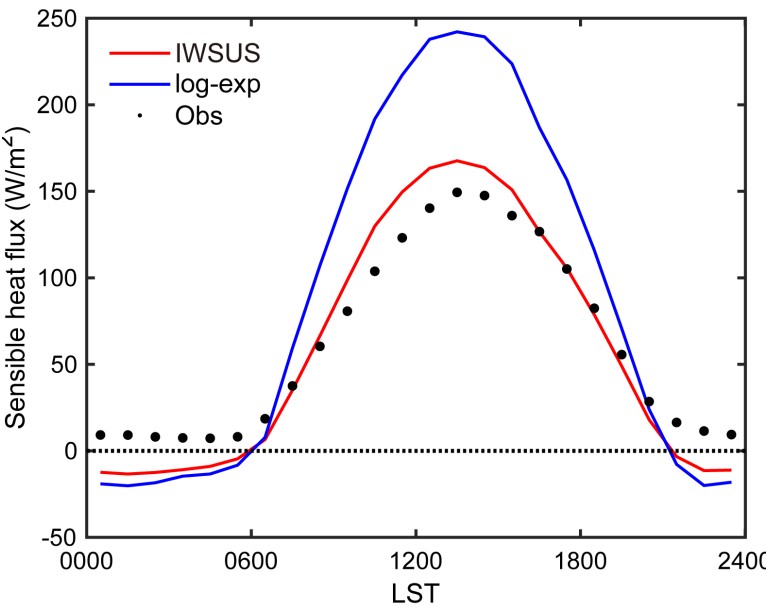


**Figure 10 Daily variation of sensible heat flux. Red line and blue line represents fluxes obtained from IWSUS and exp-log law (Masson, 2000) respectively. Dots represent the flux observation from Offerle et al. (Offerle et al., 2006).**

**5 Conclusion**

In this study, a new scheme IWSUS based on CFD simulation were developed to represent the heterogeneity of wind fields in

urban building-street canyon. By compared with exp-log scheme which is widely used in most of the current urban canopy models, the IWSUS scheme performs better in describing the spatial heterogeneity of the wind field caused by the complex and heterogeneous urban canopy structure.

1. The wind speed vertical profiles in IWSUS scheme are quite different from the traditional logarithmic-exponential wind profiles but in better agreement with the observation and CFD simulation, because the exp-log wind speed scheme assumes of

flat, open, and homogeneous land surface, while the IWSUS is based on the observation and CFD simulation of the complex and heterogeneous urban canopy structure.

2. In IWSUS scheme, the wind speed/directions and street canyon geometry (denoted by *AR*) are taken into account as the main parameters to describe the homogenous wind field distribution in and over the complex and homogenous street canyon. The wind speed vertical profiles over the different representative positions are varies in IWSUS. The result by IWSUS is in

good agreement with the observation, and its performance is significantly better than that by exp-log scheme in which the wind speed is uniform in the horizontal direction.

3. The horizontal averaged wind speed and near-surface energy flux over the urban area are simulated by the IWSUS scheme and compared with those by exp-log scheme. The evaluation by observations shows that the accuracy of the IWSUS is significantly higher than that by exp-log scheme.

Therefore, IWSUS will be suitable to be applied to the simulation of low atmospheric meteorological environmental process in the meso-scale atmospheric model.

**Code and data availability**

All of the data generated during the current study and the code of IWSUS are available:

1. The data for all the results included in this paper can be found in Zonodo doi link: https://doi.org/10.5281/zenodo.7372523;
a document named supplement.docx, in which more figures and results are included also can be found in the above link;

2. The source code for IWSUS-v0.1 and the test case can be downloaded in https://doi.org/10.5281/zenodo.7488104.

3. The CFD were used in this study for simulations of perpendicular and parallel scenarios, all of the files of model setting and simulating output are also available at https://doiora/10.5281/zenodo.7371305 and https://doi.org/10.5281/zenodo.7371804

4. The coupling codes for IWSUS-v0.1 with Weather Research and Forecasting model (WRFV3.9.1.1) and Gotherburg case can be downloaded in https://github.com/krmyArag/IWSUS.

**Author contribution**

Liu Zhenxin contributed to the conception and methodology of the study; the work of CFD simulation and parameterization were performed and provided by Chen Yuanhao; the field experiments and observation data provided by Liu Cheng; Wang
Yuhang, Liao Hong and Liu Shuhua provides key expert guidance on the initial setting of simulation experiments. The paper was written by Liu Zhenxin and Chen Yuanhao.

**Competing interests**

The authors declare they have no competing interests.

**Acknowledgements**

This work was supported by the Natural Science Foundation of Jiangsu Province (*Grants No. BK20220031*) and the National Natural Science Foundation of China (*Grants No. 41405019; 41630530*)
The authors also would like to thank the anonymous reviewers for their constructive and helpful comments and suggestions on this manuscript. We acknowledge the High-Performance Computing Centre of Nanjing University of Information Science & Technology for their support of this work.

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
