# Peer review of "The development and validation of an Inhomogeneous Wind Scheme for Urban Street (IWSUS-v1)"

_Geoscientific Model Development, 2022_

## Author Response (AR2)

**Summary**

We thank the two reviewers and chief editor for providing valuable comments on the first version of our submitted paper. We have done our best to take into account all remarks raised. In the following we give a detailed list of all the changes made in response to the points raised by the reviewers.

Thank you once more for your help in improving our paper.

**Response to Reviewer 1**

Dear Professor X.-L. Cheng,

We would like to appreciate for your careful reading and valuable comments. As you concerned, there are several problems that need to be addressed. According to your nice suggestions, we have made extensive corrections to our previous draft. Here are the point-to-point responses to your comments. Words in red are the changes we have made in the manuscript.

*1. Line 60, please give the full name of TEB.*

**Response:** TEB is the abbreviation of Town Energy Budget which is a urban canopy model developed by Masson in 2000 (Masson, 2000). The revised paper will make this clearer.

*2. Eq. 3 σi is incorrect.*

**Response:** We are really sorry for out careless mistakes. Thank you for your reminder. In our resubmitted manuscript, we have corrected $\sigma_i$ into $\sigma_k$ for representing the turbulent Prandtl number in $k$ - $\varepsilon$ equations.

*3. In section 2, please give the boundary conditions.*

**Response:** Thank you for your reminder. As you suggested, we will add a more specific description of boundary conditions in the revised paper. The wind speed in the inlet boundary is set as a constant of 0.1 m/s, 1 m/s and 8 m/s for different scenarios. The k (turbulent kinematic energy) and ε (turbulent dissipation) are determined by $k = 1.5(I\overline{u_0})^2$ and $\varepsilon = C_\mu^{3/4} k^{3/2}/l$ respectively where $I$ represents the turbulent strength and $l$ represents the turbulent

characteristic length scale. The no-Slip boundary is applied in the ground and building walls. The other boundaries are all set to be zero-gradient.

***4. In section 2.3.2, is "residence" or "resistance"?***

**Response:** We feel sorry for our carelessness again. In our revised manuscript, we will correct "residence" into "resistance".

***5. In Figure 3-5, the symbols are not clear, some symbols overlap together and can not be distinguished.***

**Response:** We agree with you. The normalized wind profiles from CFD simulation of 1 m/s and 8 m/s scenarios are so close at each representative position, which caused the sever overlap in these figures. Therefore, we remove the presentation of 1 m/s case in Figure 3-5 and changed the color, scatter points density and shape for better recognizability, as shown in Figure R1-R3. We add an explanation at Line 216 in revised paper: It is noted that the simulation results of 1 m/s and 8 m/s scenarios are very close after normalized by inflow speed, so the following figures only shows the inflow scenarios of 0.1 m/s and 8 m/s. The original results of wind profiles at different positions can be found in Zenodo links in Code and data availability section.

But the symbols in modified Figure R2c-d and Figure R3 also somewhat overlap. Figure R2 and figure R3 contains $v$ component profile results of perpendicular and parallel scenarios respectively under different inflow conditions. For perpendicular scenario, the values of $v$ component are relatively small due to the block of buildings in the middle part of street (Figure R2 c-d). For the higher levels of parallel scenario, the $v$ component approach to the inflow wind speed so the values and variation are closed after normalization (Figure R3). We also add a more specific description for Figure R3 of perpendicular scenario profile results in revised paper: The results of $u$ component show convergence in the entrance because of waking flows from building blocking while relatively small values at the middle and the exit. Due to the lack of blocking effects to $v$ component in parallel scenarios, the value of $v$ component is higher and increase rapidly with height at lower levels in street canyon, which is obvious at the entrance. As reaching to the half of building height, the increasing rate is much smaller and gradually approach to the inflow speed.

In addition, the original and processed data are all public. The original CFD simulation

setting files and result data are public at Zenodo links https://doi.org/10.5281/zenodo.7371305 and https://doi.org/10.5281/zenodo.7371804 for perpendicular scenarios and parallel scenarios respectively. The normalized data for figure 3-5 can also be downloaded from Zenodo links https://doi.org/10.5281/zenodo.7372523, which can repeat the presented figures in the manuscript.

[Figure]

Figure R1 Modified Figure 3 for u component vertical profiles of perpendicular scenarios.

[Figure]

Figure R2 Modified Figure 4 for v component vertical profiles of perpendicular scenarios.

[Figure]

Figure R3 Modified Figure 5 for v component vertical profiles of parallel scenarios.

*6. In section 3.1, the simulation results should be tested by observation results.*

**Response:** The CFD configurations in our study are commonly used in urban street micro-scale simulations which is validated by wind tunnel experiments (Hertwig et al., 2012; Ai and Mak, 2017, 2018; Huang et al., 2019; Mirzaei, 2021). Besides, it is really hard to conduct measurement experiments under various scenarios. Therefore, it is almost impossible to test the CFD simulation results by observations. On the other hand, we developed the IWSUS model based on these CFD simulation results under the tested configuration and compared the wind speed with observations in real street canyon under different heterogeneous conditions in section 4. The results show that the wind profile derived from IWSUS is more consistent with the observation, which can also demonstrate the CFD simulation is reliable.

*7. Section 3.2 is very confusing, why the expressions of ur (or vr) with AR are given, not ur (or vr) with hr?*

**Response:** We apologize for our confusing expressions in section 3.2. The expressions of $u_r$ (or $v_r$) with AR are given in order to determine the segmentation points of piecewise profile functions. The two parameters, normalized height ($h_r$) and normalized wind speed ($u_r$ or $v_r$) of every segmentation point, are needed, and only vary with AR. Therefore, these mathematical expressions are needed. Then, the type of function for every piece at different heterogeneous representative positions are specified according to the variation style, and the coefficients in these specified functions are solved by substituting the heights and wind component values at segmentation points in both ends of the piece. Finally, the expressions of wind profiles are obtained, which is also the expressions of $u_r$ (or $v_r$) with $h_r$ as concerned.

We will add the red words above, the mathematical expressions between $u_r$ (or $v_r$) and $h_r$, and their determine functions for coefficients under perpendicular scenarios in revised manuscript and under parallel scenarios in the re-arranged supplement materials.

**References**

Ai, Z. T. and Mak, C. M.: CFD simulation of flow in a long street canyon under a perpendicular wind direction: Evaluation of three computational settings, Building and Environment, 114, 293-306.https://doi.org/10.1016/j.buildenv.2016.12.032, 2017.

Ai, Z. T. and Mak, C. M.: Wind-induced single-sided natural ventilation in buildings near a long street canyon: CFD evaluation of street configuration and envelope design, Journal of Wind Engineering and

Industrial Aerodynamics, 172, 96-106.https://doi.org/10.1016/j.jweia.2017.10.024, 2018.

Hertwig, D., Efthimiou, G. C., Bartzis, J. G., and Leitl, B.: CFD-RANS model validation of turbulent flow in a semi-idealized urban canopy, Journal of Wind Engineering and Industrial Aerodynamics, 111, 61-72.https://doi.org/10.1016/j.jweia.2012.09.003, 2012.

Huang, Y.-D., Hou, R.-W., Liu, Z.-Y., Song, Y., Cui, P.-Y., and Kim, C.-N.: Effects of Wind Direction on the Airflow and Pollutant Dispersion inside a Long Street Canyon, Aerosol and Air Quality Research, 19, 1152-1171.https://doi.org/10.4209/aaqr.2018.09.0344, 2019.

Masson, V.: A physically-based scheme for the urban energy budget in atmospheric models, Boundary-layer meteorology, 94, 357-397.https://doi.org/10.1023/A:1002463829265, 2000.

Mirzaei, P. A.: CFD modeling of micro and urban climates: Problems to be solved in the new decade, Sustainable Cities and Society, 69.https://doi.org/10.1016/j.scs.2021.102839, 2021.

**Response to Reviewer 2**

Dear Referee #2,

Thank you for valuable comments. The followings are the point -to-point reply.

**-Major comments:**

***1. While I admit a heterogeneous wind field may improve the simulation of meteorological processes (lines 68-71), but the authors only focus on the neutral atmospheric conditions (neither buoyancy nor thermodynamics is accounted for in their theoretical framework), so I am questioning the validity of using the IWSUS scheme to estimate the surface energy fluxes.***

**Response:**

We quite agree with you. The vertical stability of the near ground atmosphere is indeed an important factor affecting the wind profile and the surface flux transport. But in this article, we mainly focus on how the spatial distribution of urban near-surface wind profiles change when the assumption of heterogeneous urban canopy structure is applied instead of the assumption of homogeneous underlying surface in the urban canopy model.

It should be emphasized here that the logarithmic-exponential vertical wind profile, which acts as the control group in the simulation comparison of this paper, is based on the assumption of horizontal homogeneous underlying surface. It was originally used to simulate the wind profile over the land surface of vegetation-soil type in the meso-scale models. Then it was later applied for the urban canopy land surface in the most current urban canopy models/ schemes, by modifying some key parameters, such as roughness length and zero plane displacement, to make it more coincide with the characteristics of the urban underlying surface.

Therefore, there is no need to explicitly consider the difference of the vertical wind profile in various vertical atmospheric stability conditions, such as buoyancy or thermodynamics atmospheric layer scenarios, in the log-exponential vertical wind profile model. Correspondingly, the IWSUS wind profile scheme obtained in our study, which is applied to replace the log-exponential profile scheme in the urban canopy model, have not further considering the effects of **buoyancy or thermodynamics** conditions though it is the quite important scientific issue.

Compared with the log-exponential profile scheme, the IWSUS scheme has many advantages: on the one hand, the mean wind profile and energy flux in the land surface grid in model by IWSUS are closer to the observation; on the other hand, the IWSUS

scheme can obtain the wind profile for different types of locations in the street canyon. Instead of obtaining only the average wind profile in each land surface grid in model as the log-exponential profile.

We have added the relevant discussion at Line 389 to 406 marked in the revised version.

***2. I encourage the authors to provide more complete description in Section 2.2. In this case, I think the length of the building is also a relevant controlling parameter, while the authors vary the aspect ratio H/W, the shape of the building H/Lh is also different. I am wondering if it is reasonable to take AR as the sole controlling parameter in developing the IWSUS scheme.***

**Response:**

We agree with you that the Height Length Ratio (HLR) of the building canopy may indeed affects the wind field distribution and urban land-surface flux. For example, in Figure 1b of the article, when the length of the street canyon tends to infinity, the wind speed profile of most areas in the street canyon should be close to those over points C and D, which represent the situations in the middle of the street valley. Thus the weights of these two points for calculate the average wind speed and flux in the urban grid points will be close to 1; On the contrary, when the HLR is small, the weights of points A, B, E and F in the figure will be larger. Therefore, HLR is a factor in the aerodynamic modeling in and over the urban canyon.

However, we note that most of the current urban canopy models, including Masson's TEB and WRF-UCM, do not introduce HLR as the main input parameter to describe the urban canopy geometry character. Only the building Height and street Wide Ratio (HWR) is usually used for this purpose. This performance is based on practical experience of model development: first, introducing more input parameters into a numerical model often results in greater complexity and the calculating instability. Secondly, according to our current work progress (but regrettably no published paper yet), the impact of HLR on urban canopy simulation results is not only far less than that of HWR, but also be more effective in the radiation energy balance. That's why we have not applied HLR into IWSUS in this study.

***3. Equation 6 and Figures 8 and 9: I am curious why the wind profile within the urban canopy is uniformly distributed, shouldn't the exponential wind profile be a function of z?***

**Response:**

Equation 6 is from TEB model (Masson, 2000), where $U_{can}$ does not represent the variation of wind speed with height ($z$) in the street canyon, but it is the value of wind speed at a specific height in the street canyon by TEB. The author believes that the $U_{can}$ can well represent the average wind speed in the street canyon, which is modeled as infinite long valley with various orientations. This scheme is currently used in many earth system models, such as CLMU (Community Land Model, Urban module), we were easily to obtain the corresponding model code from CLMU and compare its simulation results with these by our IWSUS scheme. Therefore, we list the corresponding equations as Equations 6, 8 and 9 in this paper.

***4. The authors didn't provide sufficient information on the boundary conditions and grid resolutions. Also, pls justify the use of such a small domain size. Given the building height H can be as much as 30m, I don't think a 50m domain height is large enough to carry out a convincing CFD simulation.***
**Response:**

Thank you for your suggestion. This is indeed a point that is worth discussing in this article.

Firstly, the CFD simulation domain in this study $L_x \times L_y \times L_z$ was set as 200 m × 100 m × 50 m, with a grid resolution of 1 m. $k$ and $\varepsilon$ at the inflow boundary are calculated from $k = 1.5(I\overline{u_0})^2$ and $\varepsilon = C_\mu^{3/4}k^{3/2}/l$, where $I$ is the turbulent intensity and $l$ is the turbulent characteristic length scale. It is a widely used methods to set the initial boundary conditions in a street canyons CFD simulation that the ground and all walls are no-slip boundary conditions, and the other outer boundaries are zero-gradient boundary conditions. The introduces of the boundary conditions setting above will be added into the revised version of the manuscript.

Secondly, as you pointed out, it is uncommon to set a domain height of only 50 m in a scenario with building height of 30 m. In most of other similar simulations, the boundary in the vertical direction is often set to be more than 6 times the height of the building. Factually at the beginning of this study, we also tried to set as the common sense, but the results did not match our research needs:

In this study, the CFD numerical experiment is set up to generate the IWSUS wind profile scheme which will be mainly applied in the meso-scale atmospheric model. In the current meso-scale model (taking WRF as an example), the wind field at the lowest atmospheric layers is the input parameters for the land surface scheme (that is, the role

position of IWSUS), and then to further calculate the wind profile near the ground. The height of the lowest free atmospheric layer is often set as about 1.5-2 times the average height of the land canopy, sometimes even just slightly higher than the average height of urban buildings.

Therefore, when we set the height of the upper boundary in CFD as the more common way, for example about 6 times the building height in our CFD model, the corresponding parameterized wind vertical profiles in IWSUS were not match the physics pictures of the mesoscale model in which the IWSUS is coupled into. On the contrary, when the height of the upper boundary in the CFD simulation domain is set to 50 m while the highest building height is 30 m, the results is more matched with the scenarios of the lowest layer of the free atmosphere and the urban canopy in the mesoscale model. This unusual setting better ensures the consistency between the CFD scenarios and the requirements in the mesoscale model. In addition, previous studies on CFD simulation of street canyons (Martilli and Santiago, 2006; Santiago et al., 2006; Santiago et al., 2008; Yang and Shao, 2008; Cui et al., 2019; S Sützl et al., 2021) also support the rationality of this approach.

This information has been added to the manuscript: *In meso-scale atmospheric models, the lowest layer of free atmosphere is often set as about 1.5-2 times the average height of urban canopy. Since IWSUS will be mainly applied in the meso-scale models, the vertical height of the domain is set as 50 m in our CFD experiments to match with the requirements in meso-scale model.*

**-Minor comments:**

*1. Section 2.1, line 85, "The CFD method was applied in study to analyzes", grammatical mistake.*

**Response:** Thanks for pointing out the grammatical mistake. "analyzes" has been corrected to "analyze" in the next version of the manuscript.

*2. Section 2.1, line 94, "p\* is a modified mean kinematic pressure", grammatical mistake.*

**Response:** Thanks for pointing out the grammatical mistake. "a" has been corrected to "the".

*3. Section 2.3, lines 135 and 136, the initial letters should be capitalized.*

**Response:** Thank you very much. The problem has been corrected.

*4. Section 2.3, line 138, "Masson (Masson, 2000)", the literature is inappropriately cited.*

Response: Thank you very much. The problem has been corrected.

*5. Section 2.3, line 140, "In the exp-log wind profile scheme", some relevant references are needed.*

Response: We are sorry for our unclear expression. The exp-log wind profile scheme here is the scheme developed by Masson as mentioned in the former part of this paragraph. We will improve our expression in a clearer way in the next version of the manuscript.

**Reference**

[1]  Cui, D., Hu, G., Ai, Z., Du, Y., Mak, C. M., and Kwok, K.: Particle image velocimetry measurement and CFD simulation of pedestrian level wind environment around U-type street canyon, Building and Environment, 154, 239-251.https://doi.org/10.1016/j.buildenv.2019.03.025, 2019.

[2]  Martilli, A. and Santiago, J. L.: CFD simulation of airflow over a regular array of cubes. Part II: analysis of spatial average properties, Boundary-Layer Meteorology, 122, 635-654.https://doi.org/10.1007/s10546-006-9124-y, 2006.

[3]  Masson, V.: A physically-based scheme for the urban energy budget in atmospheric models, Boundary-layer meteorology, 94, 357-397.https://doi.org/10.1023/A:1002463829265, 2000.

[4]  Santiago, J. L., Coceal, O., Martilli, A., and Belcher, S. E.: Variation of the Sectional Drag Coefficient of a Group of Buildings with Packing Density, Boundary-Layer Meteorology, 128, 445-457.https://doi.org/10.1007/s10546-008-9294-x, 2008.

[5]  Santiago, J. L., Martilli, A., and Martín, F.: CFD simulation of airflow over a regular array of cubes. Part I: Three-dimensional simulation of the flow and validation with wind-tunnel measurements, Boundary-Layer Meteorology, 122, 609-634.https://doi.org/10.1007/s10546-006-9123-z, 2006.

[6]  Sützl, B. S., Rooney, G. G., and van Reeuwijk, M.: Drag distribution in idealized heterogeneous urban environments, Boundary-Layer Meteorology, 178, 225-248, 2021.

[7]  Yang, Y. and Shao, Y.: Numerical simulations of flow and pollution dispersion in urban atmospheric boundary layers, Environmental Modelling & Software, 23, 906-921.https://doi.org/10.1016/j.envsoft.2007.10.005, 2008.

**Response to Chief Editor Comments**

Dear Editor Astrid Kerkweg,

Thank you for your valuable comments. Our responds are as follows:

*1. The main paper must give the model name and version number (or other unique identifier) in the title.*

**Response:**

We have added a version number in the title as you suggested in the first comment, changing the title into "The development and validation of an Inhomogeneous Wind Scheme for Urban Street (IWSUS-v1)". the revision can be found in the next version of our manuscript.

*2. If the model development relates to a single model then the model name and the version number must be included in the title of the paper. If the main intention of an article is to make a general (i.e. model independent) statement about the usefulness of a new development, but the usefulness is shown with the help of one specific model, the model name and version number must be stated in the title. The title could have a form such as, "Title outlining amazing generic advance: a case study with Model XXX (version Y)".*

**Response**:

There are a cluster of cases simulation conducted via OpenFOAM in our research aims to provide various vertical wind profiles of different scenarios. Those works offered really necessary data support for our IWSUS development but the main point of this work is **only the IWSUS development**. In other word, we do NOT think the main work we did was a case study work of OpenFOAM. Our main purpose is to conduction a more reasonable and realizable parameterization scheme for the wind field distribution in and over a regularly arranged building-street canyon. Both the observations and the simulations (It happened that we used OpenFOAM to finish our simulating cases) were applied in our work to complete this idea. It would still make sense if we configured the simulating cases in any other aerodynamical fluid tools beside the OpenFOAM to provide the corresponding wind field simulation results with different scenarios of initial AR and background wind speed. Therefore, we do NOT think our study is only a case study with OpenFOAM. And we think it is not probable either to mention the word OpenFOAM in the title. It may cause some misunderstand to readers the main idea of this article.

Thanks very much for your carefully concern to our manuscript.